# Modeling the shared risks of malaria and anemia in Rwanda

**Pacifique Karekezi**[1]*, **Jean Damascene Nzabakiriraho**[2], **Ezra Gayawan**[1,3]

**1** African Institute for Mathematical Sciences (AIMS), Kigali, Rwanda, **2** African Centre of Excellence in Data Science, University of Rwanda, Kigali, Rwanda, **3** Department of Statistics, Federal University of Technology, Akure, Nigeria

☯ All these authors are contributed equally to this work.
* pacifique.karekezi@aims.ac.rw

**Data Availability Statement:** All relevant data are within the paper and its Supporting information files.

**Funding:** The authors received no specific funding for this work.

## Abstract

In sub-Saharan Africa, malaria and anemia contribute substantially to the high burden of morbidity and mortality among under-five children. In Rwanda, both diseases have remained public health challenge over the years in spite of the numerous intervention programs and policies put in place. This study aimed at understanding the geographical variations between the joint and specific risks of both diseases in the country while quantifying the effects of some socio-demographic and climatic factors. Using data extracted from Rwanda Demographic and Health Survey, a shared component model was conceived and inference was based on integrated nested Laplace approximation. The study findings revealed similar spatial patterns for the risk of malaria and the shared risks of both diseases, thus confirming the strong link between malaria and anaemia. The spatial patterns revealed that the risks for contracting both diseases are higher among children living in the districts of Rutsiro, Nyabihu, Rusizi, Ruhango, and Gisagara. The risks for both diseases are significantly associated with type of place of residence, sex of household head, ownership of bed net, wealth index and mother's educational attainment. Temperature and precipitation also have substantial association with both diseases. When developing malaria intervention programs and policies, it is important to take into account climatic and environmental variability in Rwanda. Also, potential intervention initiatives focusing on the lowest wealth index, children of uneducated mothers, and high risky regions need to be reinforced.

## Introduction

In sub-Saharan Africa, malaria and anemia contribute substantially to the high burden of morbidity and mortality among under-five children. In Rwanda, both diseases have remained as public health challenge over the years in spite of the numerous intervention programs and policies put in place. The well-being and survival chances of women and young children have continued to be endangered by the diseases [1, 2]. The occurrence and magnitude of other diseases and malnutrition are exacerbated by malaria parasite leading to severe morbidity and mortality in some extreme cases. Malaria parasites can consume iron molecules in an individual's red blood cell causing low functional hemoglobin concentration. In the case of young

**Competing interests:** The authors have declared that no competing interests exist.

children, low hemoglobin concentration results in severe anemia and prevents quick recovery [3]. Anemia, commonly caused by iron deficiency and this occurs in situation where there is insufficient amount of iron in the red blood cells [4]. Malaria parasites feed on iron in the blood cell, leading to shortage of hemoglobin (Hb) concentration level. Thus, there are synergies between the causal relationship of malaria and anemia. *Plasmodium falciparum* can encroach into the red blood cell leading to acute hemolysis thereby hindering the proper development of the red cell, causing severe anemia [5]. On the other hand, severe anemia worsens morbidity from malaria particularly in children [6].

In both the tropics and subtropics, anemia is considered a health risk for under-five children particularly for those who simultaneously suffer from malaria [7].

It is a condition in which the blood is in short of healthy red blood cells, which could be due to malfunctioning of existing red blood cells that affects the flow of oxygen to the body tissue [8]. Under-five children and pregnant women are most affected by the global public health issue; with the World Health Organization estimating that 42% of children under this group and 40% of pregnant women are anemic worldwide [9]. Children who suffer from anemia experience increased heart rate, poor wound and tissue healing, poor cognitive development all of which compromise the child's quality of life [10].

Malaria and anemia put direct pressure on the already fragile health care systems of developing countries, and also negatively affect individuals and households because of the loss of income and other economic consequences that arise due to loss of man-hour occasioned by the time spent during the recovery process or in taking care of the sick one [2]. Many developing countries are malaria endemic, where a huge proportion of the annual documented 300 million clinical cases and 2 million deaths take place. For instance, in 2019, 95% of the 229 million reported malaria cases came from 29 developing, while malaria deaths were put at 409,000 and about 67% of this number occurred among under-five children [11].

Sub-Saharan African countries jointly account for about 93% of all malaria cases and 94% of malaria deaths that took place in 2018 with majority of the death reported among under-five children [9]. This is in spite of the multiple evolving government policies coupled with interventions from international development agencies. In Rwanda, malaria is the second leading cause of morbidity and is responsible for about 7.4% per cent of outpatient hospital visits and 4.3% of malaria proportional mortality [12]. Prior to year 2012, the country experienced a downward trend in malaria cases culminating to a record low of 48 per 1,000. However, between this period and 2016, there was a reversal in the gains that saw the number of reported cases rising to about 403 per 1,000 in 2016 [13].

Consequently, malaria gains in the country can be considered fragile and sustained policies that can push back new cases need to be scaled up. Human activities and behaviour, climatic change, environmental modification, and insecticide resistance play important roles in the sudden surge in cases [13, 14]

The incidence of malaria has been linked to altitude, with a higher prevalence in lowlands than in highlands settings, and more cases occur around May-June and November-December. In addition to favorable climate, closeness to marshlands, irrigation schemes, and movement of people within and across geographical border influence the transmission, particularly in the Southern and Eastern fringes of the country [14, 15]. As earlier argued, higher malaria risk in a given location would also imply higher risk of anemia particularly in children.

Several studies have been undertaken to evaluate the incidence and determinants of malaria and anaemia among under-five children in Rwanda including their geographical distributions [1, 14, 16–18]. However, limited efforts have gone into modeling the possible co-morbidity of the two disease, and there exists a persistent knowledge gap in their spatial overlap that specifically highlight districts with high and low risks of the combined diseases. Moreover, only few

studies have mapped malaria and anaemia prevalence while controlling for climatic conditions particularly temperature and rainfall in the country. It is important to understand the patterns of the spatial overlap of both (shared) and specific risks from malaria and anaemia at small scale levels across the country, and to quantify the dynamics of climatic conditions for the two diseases as these could effectively aid intervention programs that are aimed at addressing their occurrence among young children. Consequently, this study was designed to estimate the shared and specific spatial patterns of malaria and anemia among children under five years of age in Rwanda using data extracted from the 2014–2015 Rwanda Demographic and Health Survey (RDHS). We use a shared-component modeling approach within a Bayesian framework allowing for simultaneous estimation of metrical variables as nonlinear effects, location-specific random effects that account for spatial heterogeneity, and the usual linear effects of categorical variables. The model enables the distinct estimation of the underlying risk surface for the diseases in the form of a component shared by both diseases (refered to as the shared-component) and others that are specific to each disease (disease-specific components). The shared spatial component can be viewed as a surrogate for unobserved covariates that influence the spatial structure of both diseases. We considered the effects of temperature and rainfall as climatic in addition to other demographic factors on the shared risks of the two diseases. To produce approximations to the posterior marginals of all parameters of interest, we adopt the integrated nested Laplace approximations (INLA) approach.

## Materials and methods

### Data

The study used data obtained through the 2014–2015 Rwanda Demographic and Health Survey (RDHS). The survey, implemented by the National Institute of Statistics of Rwanda in partnership with the Rwanda Biomedical Center (RBC) and Rwanda Ministry of Health (MoH) between November 2014 and April 2015, provide essential data for quantifying and monitoring important demographic and health indicators at the national level, the five provinces, each of Rwanda's 30 districts and for urban and rural areas. The survey was executed through a two-stage sampling design where enumeration areas (EAs), referred to as clusters, were selected at the first stage from the sampling frame used during the 2012 Rwanda Population and Housing Census (RPHC). A total of 492 clusters comprising 113 in urban and 379 in rural areas were selected. The second stage involves the selection of households from the listed EAs, which were picked at random. A total of 12,793 households were selected comprising of about twenty-six households from each EA.

Eligible women for interview were those aged 15 to 49 years who were the permanent residents of the selected households or those who slept in the households the night before the survey. There were 13,564 eligible women in the selected households and 13,497 were successfully interviewed, yielding a response rate of 99.5 percent. Information was also collected on children younger than five years of age from the caregivers in all the households. Anemia and malaria testing were executed in the sub-sample of households not selected for the male survey. After informed consent was obtained from the caregiver, children in these households were tested for malaria and anemia. The malaria test was executed through the rapid diagnostic test (RTD) and thick and thin blood smears. A drop of blood was taken by pricking the end of the finger in the case of RDTs and this was also used for anemia testing.

The response variables of interest were created from the malaria and anemia indicators with each variable taking the value of one if the child suffers from the ailment and zero if otherwise.

The independent variables considered include type of place of residence, sex of a household head, bed-net ownership, bed-nets usage, household wealth index, mother's educational attainment, child's sex, child's age, and age of household head. The district of residence was geo-referenced. Furthermore, climate-related factors including temperature and rainfall were obtained from the GPS data set of the 2014–2015 RDHS to examine their effect on malaria and anemia co-morbidity. We extracted the shapefile of Rwanda from the Spatial Data Repository put together by The DHS Program [19]. This was used in constructing the adjacency matrix for the districts of Rwanda used in our Bayesian model and for plotting the estimates on maps.

### Ethics statement

The study used a secondary data obtained from the Demographic and Health Survey after approval was secured. It should be noted that DHS adopts a coordinate displacement process to ensure the preservation of the identity of the respondents. Urban clusters are displaced on a distance of up to two kilometers and up to five kilometers for rural clusters. However, this would have minimal effect on our analysis since we used an area-level spatial approach that considers the districts of Rwanda hence every sampled individual would belong to a district notwithstanding the displacement of the coordinate.

### Statistical analysis

**Shared component model.**   The shared components models described by Knorr and Best [20] allows for the estimation of the shared pattern that describes the clustering of two or more diseases and the ones specific to each disease. Consider a response variable $y_{ijd}$, taking value of 1 if a child $i$ from district $j$ is suffering from disease $d$, and 0 otherwise, where, $i = 1, 2, \cdots,$ 3446, $j = 1, 2, \cdots,$ 30, and $d = 1, 2$. We assumed that the outcome $y_{ijd}$ has a Bernoulli distribution denoted as

$$y_{ijd} \sim \text{Bernouilli}(p_{ijd}),\tag{1}$$

where

$$f(y_{ijd}|p_{ijd}) \quad = p_{ijd}^{y_{ijd}}(1 - p_{ijd})^{1-y_{ijd}}\tag{2}$$

$$= \exp[y_{ijd}\eta_{ijd} - \log(1 + \exp(\eta_{ijd}))],\tag{3}$$

and $p_{ijd} = P(y_{ijd} = 1)$ and $\eta_{ijd}$ is the model predictor that describes the experience for child $i$ who lives in district $j$, given that $\eta_{ijd} = \text{logit}(p_{ijd})$. Consequently, with the logit link function, the predictor can be linked to covariates, which are of different forms, as follows:

$$\text{logit}(p_{ijd}) = \alpha + \omega\beta + g(\nu_j) \cdot \sigma_d + g_d(s_j),\tag{3}$$

where $\alpha$ denotes the intercept common to the two diseases, $\beta$ is a vector of parameters for categorical variables shared by the two diseases, $\omega$ represents a vector that collects together all the categorical variables, $g(\nu_j)$ is the common spatial field for all diseases, $\sigma_d$ is the loadings component which controls how disease $d$ is affected by the shared field, and $g_d(s_j)$ is the disease-specific spatial components.

We implemented the model through a Bayesian inference that relies on the integrated nested Laplace approximation (INLA) as proposed by [21]. The prior distribution for $\sigma_d$ was log-normal with a mean of 0 and accuracy of 0.1. A weakly informative Gaussian prior with small precision $\tau_\beta$ on the identity matrix was considered, and used for the fixed parameters and constant term, that is $\beta \sim N(0, \tau_\beta I)$.

The intrinsic conditional auto-regressive (ICAR) prior was considered to model the shared and specific spatial random components with precisions $\tau_v$ and $\tau_s$ respectively. The approach induces spatial auto-correlation through the adjacency structure of the district. The ICAR prior for $s = (s_1, \ldots, s_n)^T$ is given in its simplest form as

$$\pi(s|\tau) \sim \frac{1}{Z_n(\tau)} \exp\left( -\frac{\tau}{2} \sum_{i \sim j} (s_i - s_j)^2 \right), \tag{5}$$

where $i \sim j$ refers to the collection of all pairs of neighbours, with two districts considered neighbours if they share a common boundary. The term $Z_n(\tau)$ is the normalizing constant while $\tau$ is a precision parameter that determines the amount of smoothness of the spatial components. We further assigned hyperpriors to the precision parameters based on penalized prior (PC prior), which is a unified prior definition for complicated models that penalizes the model's hierarchical structure. The PC prior is based on a type-2 Gumbel distribution with parameters $\tau$ and $\lambda$ set to 1 and 0.01 respectively [22]. Scaling of the hyperpriors is required to ensure that all spatial components have the same degree of smoothness.

Data preparation and frequency tables were generated using Stata 14 while the Bayesian model was implemented using the R-INLA package.

## Results

Table 1 presents the descriptive analysis of the co-morbidity of malaria and anemia based on household characteristics of the under-five children. Overall, about 49% (1673/3446) of the children had malaria, anemia, or both. About 7.5% (259/3446) had malaria, 35.6% (1228/3446) had anemia, and another 5.4% (186/3446) suffered both malaria and anemia. Among the under-five children from rural areas, a greater proportion had malaria 8.7%, (239/2738), 36.9% (1009/2738) suffered anemia whereas 6.2% (169/2738) had both malaria and anemia. Among the children from female headed households, 10.4% (74/709) suffered malaria while 7.8% (55/709) had anemia and malaria. On other hand, slightly high proportion (35.7%, 976/2737) of children from male-headed households suffered anemia when compared with with female-headed households (35.5%, 252/709). Among children from households with no bed nets, 9.9% (53/538) suffered malaria, 37.7% (203/538) had anemia, and 8.6% (46/538) suffered both malaria and anemia. The proportion of the children who suffered malaria, anemia, or both illnesses varies with the use of bed nets such that 9.5% (27/284) of children from households in which some slept under bed nets suffered malaria, whereas 7.8% (75/967) of children who did not sleep under bed nets suffered both malaria and anemia. Children from the richest households were less likely to suffer the episodes of both malaria and anemia (0.5%, 3/627). Contrarily, among children from poorest households, about 10.1% (86/852) suffered both malaria and anemia. Children whose mothers attained at least primary education are less likely to suffer malaria, anemia, or both. The study findings also show that the proportion of children who had malaria, or both malaria and anemia did not vary with child's sex. However, slightly high proportion of male children had anemia (36.7%, 645/1757).

Table 2 presents results of the linear effects for malaria and anemia showing the posterior means, standard deviations, and 95% credible intervals. When compared with children from rural areas, those from urban settings are less likely to develop both diseases, though the estimate is not significant. Children from male-headed households are less likely to have both diseases compared with those from female-headed households. Whereas ownership of bed nets in households shows lower likelihood for contracting both diseases, the estimates for usage are not significant. The study findings revealed an association between wealth quintile and coexistence of the two diseases such that children from at least the poorer households are less likely

**Table 1. Demographic characteristics of respondents, source: RDHS 2014–2015.**

| Parameter | Malaria | Anaemia | co-morbidity | Total |
|---|---|---|---|---|
| **Place of residence** | | | | |
| Rural | 239(8.7) | 1009(36.9) | 169(6.2) | 2738 |
| Urban | 20(2.8) | 219(30.9) | 17(2.4) | 708 |
| **Sex of a household head** | | | | |
| Female | 74(10.4) | 252(35.5) | 55(7.8) | 709 |
| Male | 185(6.8) | 976(35.7) | 131(4.8) | 2737 |
| **Bed-net ownership** | | | | |
| No | 53(9.9) | 203(37.7) | 46(8.6) | 538 |
| Yes | 206(7.1) | 1025(35.3) | 140(4.8) | 2908 |
| **Bed-nets usage** | | | | |
| No child uses bednet | 87(9.0) | 357(36.9) | 75(7.8) | 967 |
| All use bednets | 114(9.1) | 766(34.9) | 91(4.2) | 2195 |
| Some of them use bednets | 27(9.5) | 105(36.9) | 20(7.0) | 284 |
| **Wealth Index** | | | | |
| Poorest | 109(12.8) | 341(40.0) | 86(10.1) | 852 |
| Poorer | 66(8.9) | 277(37.7) | 48(6.5) | 735 |
| Middle | 47(7.1) | 243(36.8) | 35(5.3) | 660 |
| Richer | 30(5.2) | 182(31.8) | 14(2.5) | 572 |
| Richest | 7(1.1) | 185(29.5) | 3(0.5) | 627 |
| **Mother's education level** | | | | |
| No education | 67(9.3) | 270(37.3) | 53(7.3) | 723 |
| Primary | 179(7.8) | 813(35.3) | 122(5.3) | 2304 |
| Secondary/ higher | 13(3.1) | 145(34.6) | 11(2.6) | 419 |
| **Child-sex** | | | | |
| Female | 121(7.2) | 583(34.5) | 87(5.2) | 1689 |
| Male | 138(7.9) | 645(36.7) | 99(5.6) | 1757 |
| **Total** | 259(7.5) | 1228(35.6) | 186(5.4) | 3446 |

to contract both diseases when compared with those from the poorest households. Children whose mothers attained primary level of education are less likely to suffer from both diseases when compared with those with no education but the result for secondary/higher education is not significant. The findings on the sex of the children show non-significant effects.

The results for the spatial effects presented in Figs 1 to 12, show the maps for the posterior means, standard deviations, and 95% credible intervals, respectively for the shared effects of malaria and anemia, specific-effects for malaria and for anemia, with the results displaying uneven distributions of the burden of the diseases across the districts of Rwanda. The findings from the shared random effects (Figs 1 to 4) reveal that the highest risks of contracting malaria and anemia can be found among children from Rutsiro, Nyabihu, Ruhango, Gisagara, and Rusizi districts, while those from Rubavu, Karongi, Ngororero, Kayonza, Gatsibo, Rwamagana, Gasabo and Rulindo have moderate risks. The joint effects of Malaria and anemia seem to be negligible across the other districts of the country. The standard deviation and credible intervals maps show that there is more uncertainty about the risk of malaria and anemia in various parts of the country, particularly in the districts of Nyagatare, Rubavu, Ruhango and Nyamasheke.

In terms of the specific components, the spatial distributions for malaria (Figs 5 to 8) reveal a pattern that is somewhat similar to the shared spatial pattern obtained for the combined

**Table 2. Estimated posterior means with 95% credible intervals for the linear effects.**

| Parameter | Mean | sd | 0.025quant | 0.975quant |
|---|---|---|---|---|
| **Place of residence** | | | | |
| Rural | 1.000 | | | |
| Urban | 0.897 | 0.102 | 0.733 | 1.095 |
| **Sex of a household head** | | | | |
| Female | 1.000 | | | |
| Male | 0.758 | 0.066 | 0.666 | 0.862 |
| **Bed-net ownership** | | | | |
| No | 1.000 | | | |
| Yes | 0.588 | 0.105 | 0.478 | 0.720 |
| **Bed-nets usage** | | | | |
| No child uses bednet | 1.000 | | | |
| All use bednets | 0.969 | 0.094 | 0.807 | 1.168 |
| Some of them use bednets | 1.153 | 0.133 | 0.887 | 1.497 |
| **Wealth Index** | | | | |
| Poorest | 1.000 | | | |
| Poorer | 0.737 | 0.083 | 0.626 | 0.866 |
| Middle | 0.673 | 0.088 | 0.565 | 0.800 |
| Richer | 0.582 | 0.099 | 0.479 | 0.707 |
| Richest | 0.492 | 0.132 | 0.379 | 0.636 |
| **Mother's education level** | | | | |
| No education | 1.000 | | | |
| Primary | 0.744 | 0.066 | 0.654 | 0.847 |
| Secondary/ higher | 0.887 | 0.119 | 0.702 | 1.118 |
| **Child-sex** | | | | |
| Female | 1.000 | | | |
| Male | 0.936 | 0.057 | 0.838 | 1.047 |

diseases. Specifically, the risks for malaria are highest among children living in Rutsiro, Nyabihu, Ruhango, Rusizi and Gisagara districts but moderate for those residing in Gatsibo, Kayonza, Rwamagana, Gasabo, Rulindo, Karongi, Ngororero, and Rubavu. However, the risks are lowest among those children living in Gakenke, Nyagatare, and Kirehe districts. The standard deviation and credible interval maps reflect high uncertainty in the districts of Rubavu, Kirehe, Nyagatare, Rusizi, Nyamasheke, Nyaruguru and Gisagara. In the case of anemia (Figs 9 to 12), the risks are higher among the children living in Rutsiro, Nyabihu, Rulindom, Ngoma, and Gisagara, followed by the districts of Rusizi, Huye, Nyanza, Ruhango, Ngororero, Gakenke, Gasabo, Nyarugenge, Kicukiro, and Kayonza. However, the risks for anemia are lower in the districts of Nyamasheke, Nyamagabe, Karongi, Muhanga, Kamonyi, Bugesera, Rwamagana, Gatsibo, Gicumbi, Nyagatare, Burera, Musanze and Rubavu. Nyaruguru and Kirehe.

The nonlinear effects of child's age, age of household head, annual precipitation and mean temperature for the shared effects are presented in Figs 13 to 16 while those for the specific effects of malaria and anemia are presented in Figs 17 to 24 respectively. The Figures show the posterior means (middle lines) surrounded by the 95% credible intervals. The findings show a negative relationship between the likelihood of suffering from both diseases and the age of the child, indicating that as a child gets older, the likelihood of suffering from both diseases decreases. The estimates for household head show higher likelihood among children whose

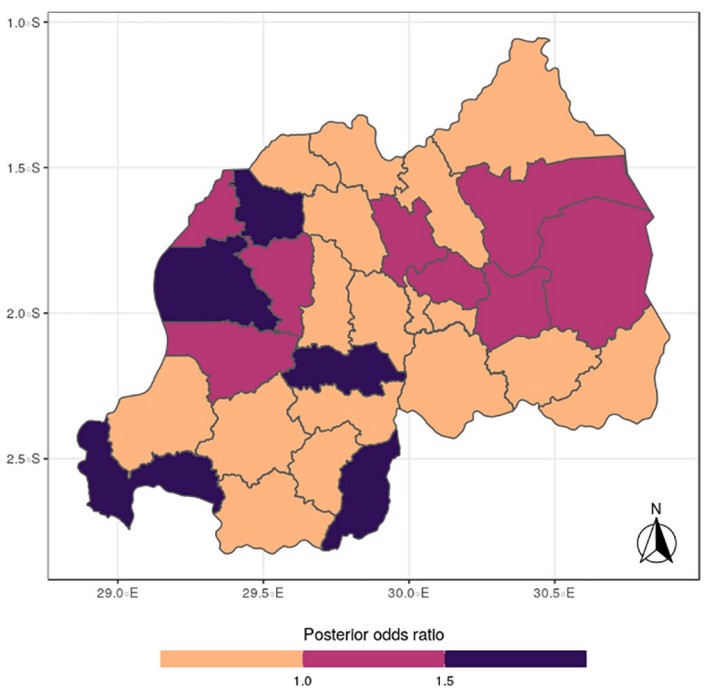

**Fig 1. Posterior mean (Source: Authors' creation).**

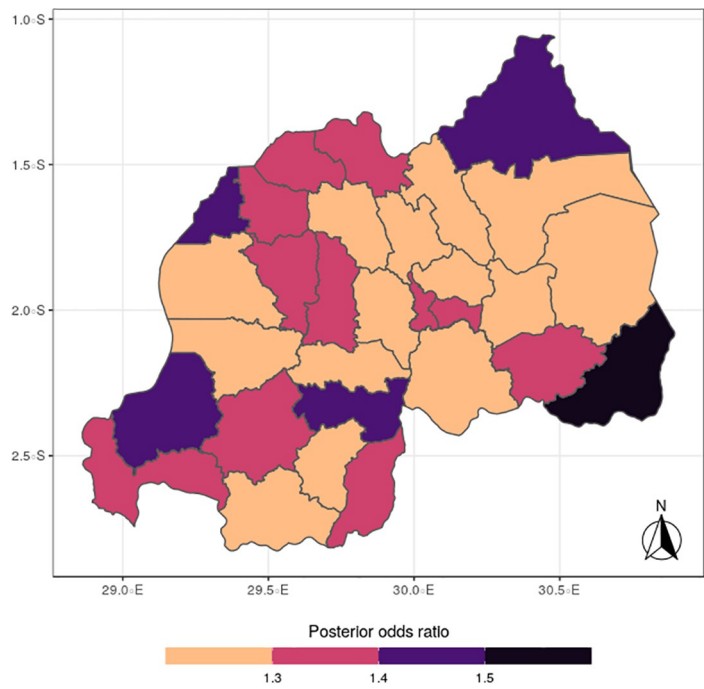

**Fig 2. Standard deviation (Source: Authors' creation).**

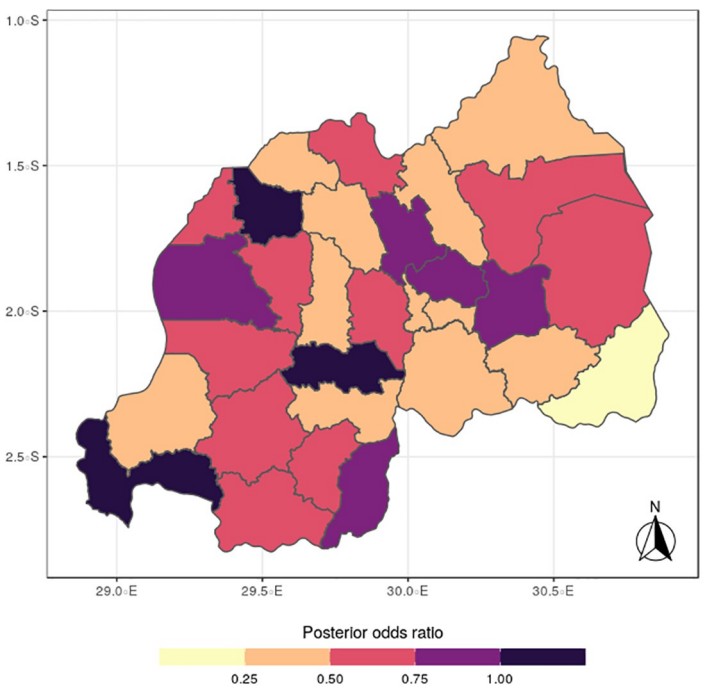

**Fig 3. (Source: Authors' creation).**

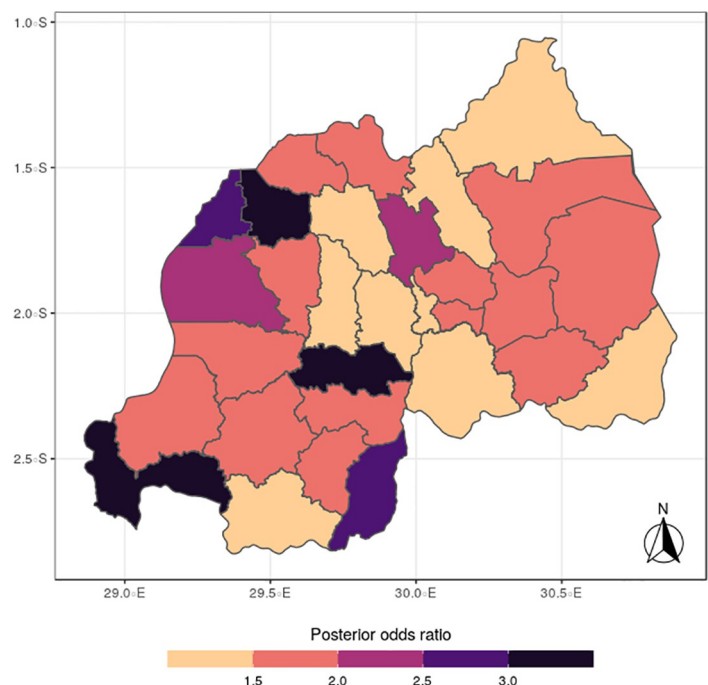

**Fig 4. 95% credible intervals for the shared effects of malaria and anemia (Figs 3 & 4) (Source: Authors' creation).**

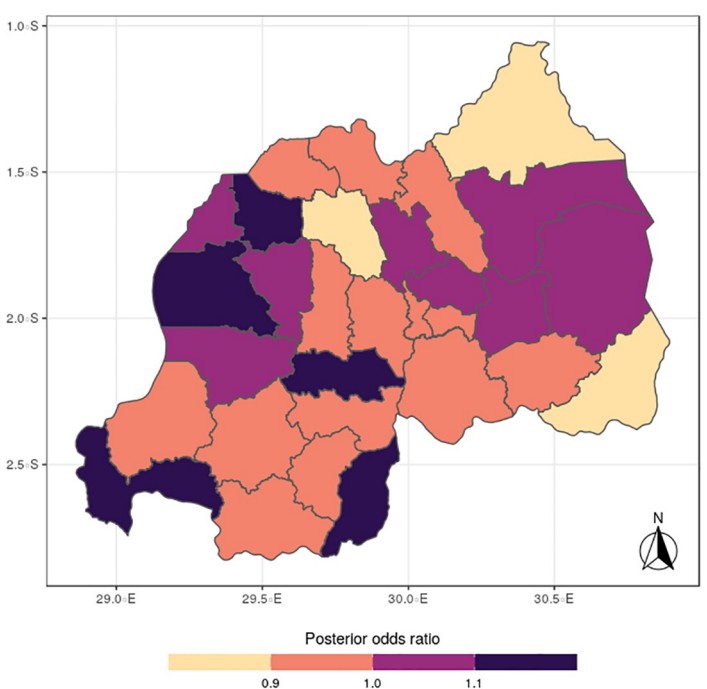

**Fig 5. Posterior mean of malaria (Source: Authors' creation).**

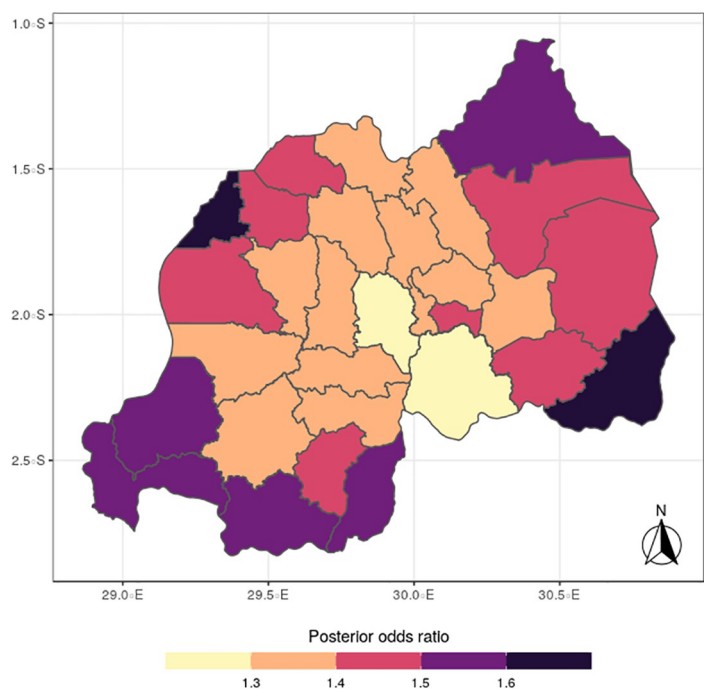

**Fig 6. Standard deviation of malaria (Source: Authors' creation).**

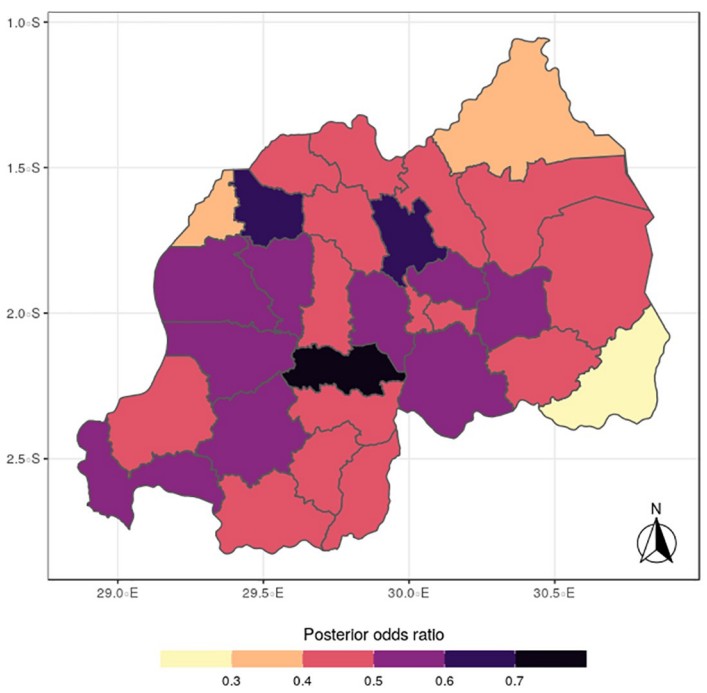

**Fig 7. (Source: Authors' creation).**

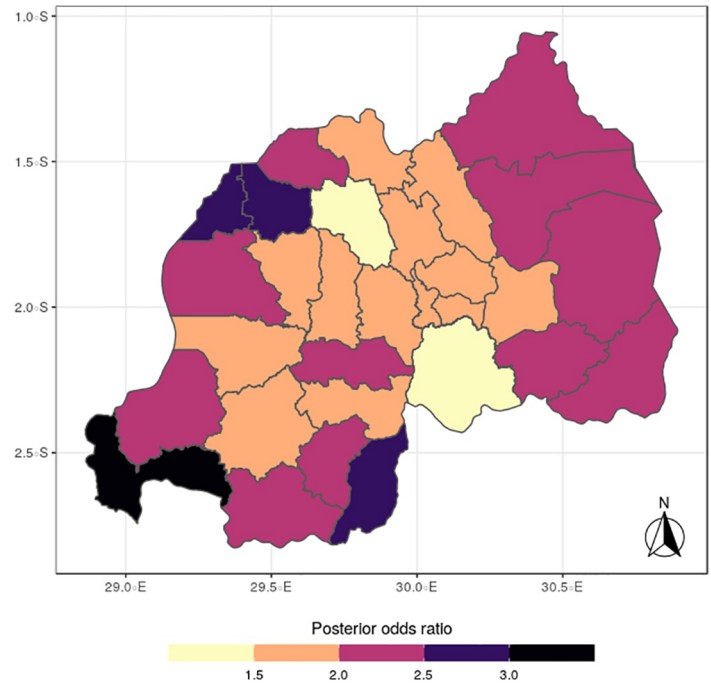

**Fig 8. 95% credible intervals for the shared effects of malaria and anemia (Figs 7 & 8) (Source: Authors' creation).**

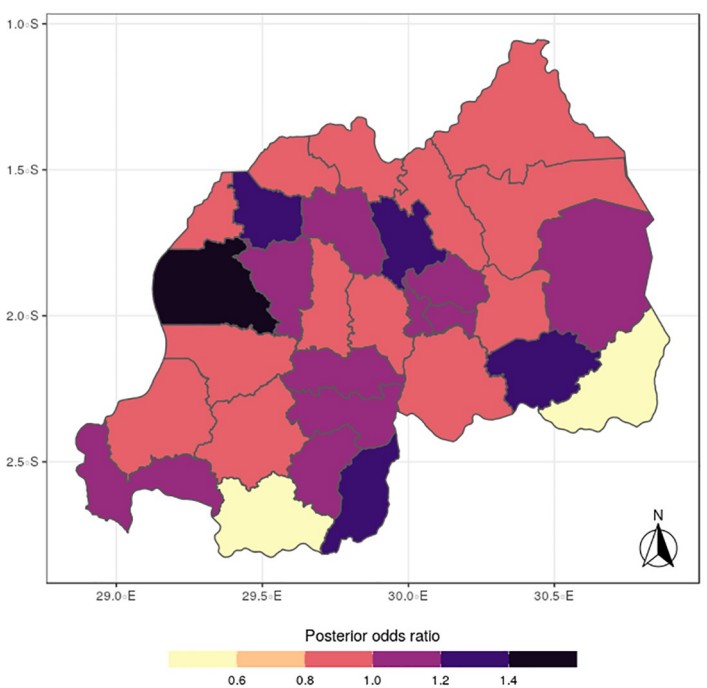

**Fig 9. Posterior mean of anemia (Source: Authors' creation).**

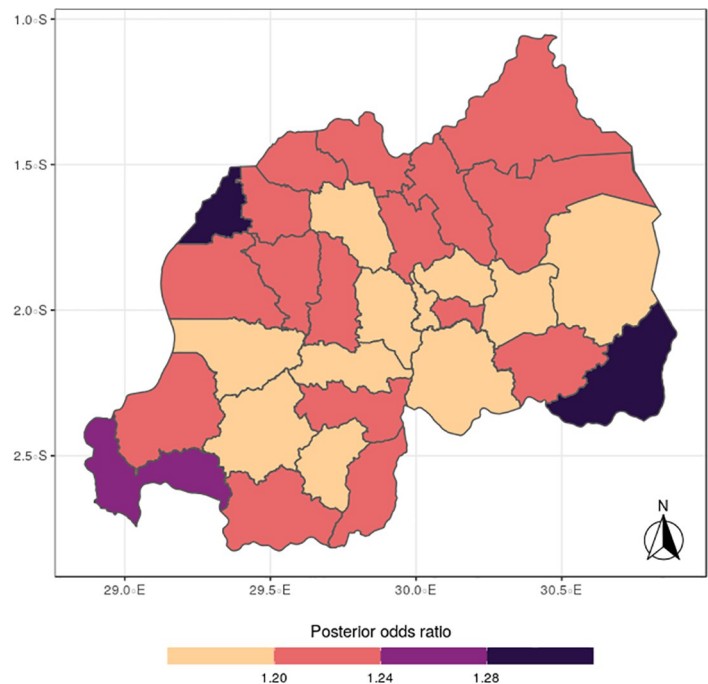

**Fig 10. Standard deviation of anemia (Source: Authors' creation).**

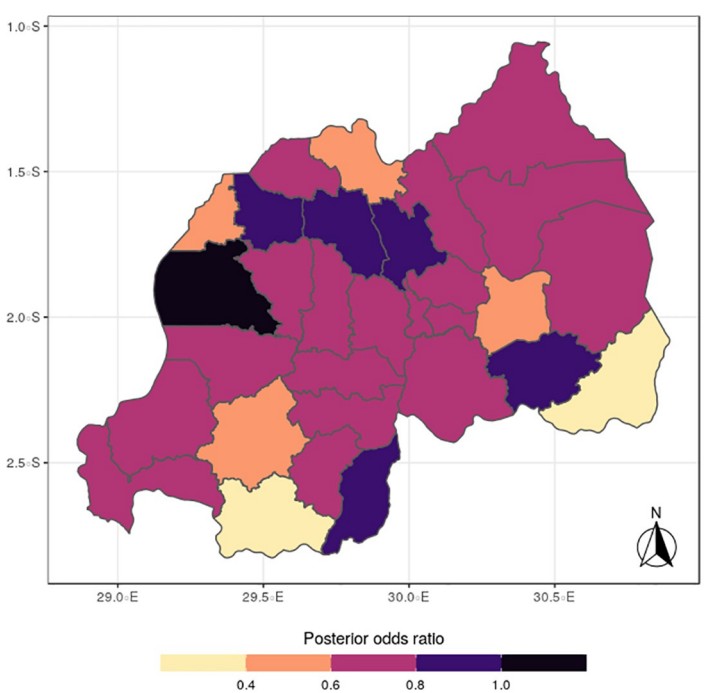

**Fig 11. (Source: Authors' creation).**

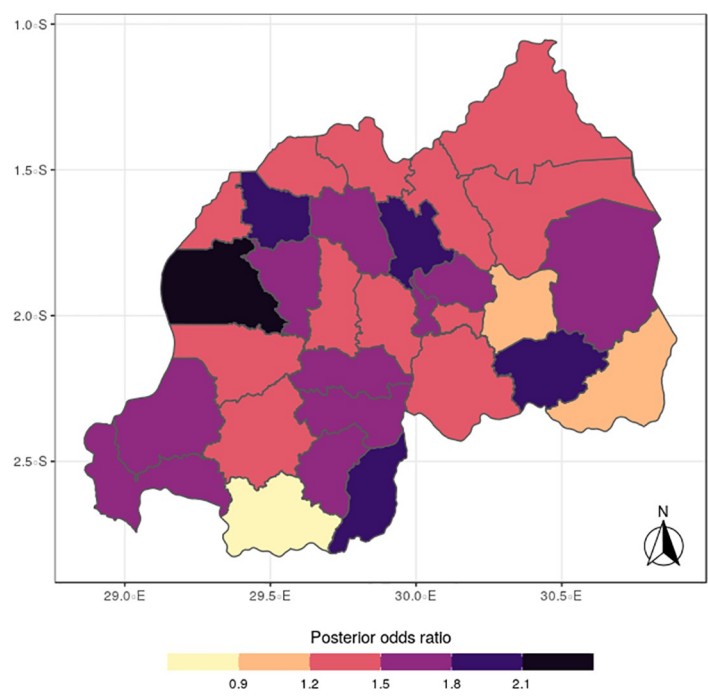

**Fig 12. 95% credible intervals for the shared effects of malaria and anemia (Figs 11 & 12) (Source: Authors' creation).**

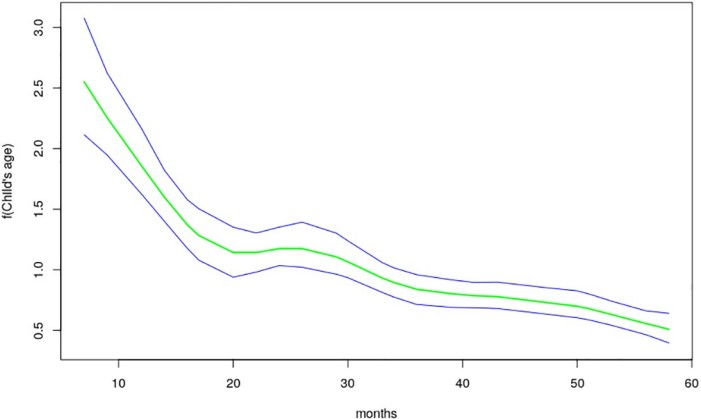

**Fig 13. Non-linear effects for the shared component of malaria and anaemia, child age (Source: Authors' creation).**

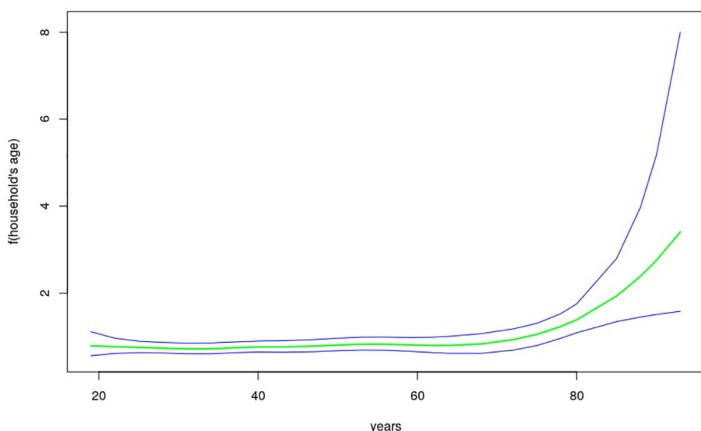

**Fig 14. Non-linear effects for the shared component of malaria and anaemia, household age (Source: Authors' creation).**

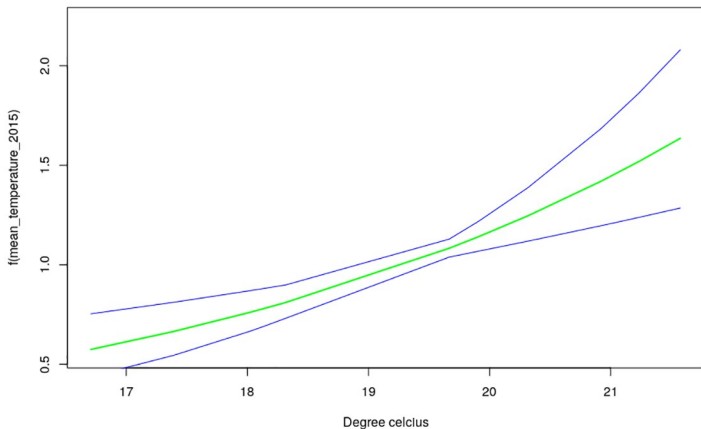

**Fig 15. Non-linear effects for the shared component of malaria and anaemia, mean temperature (Source: Authors' creation).**

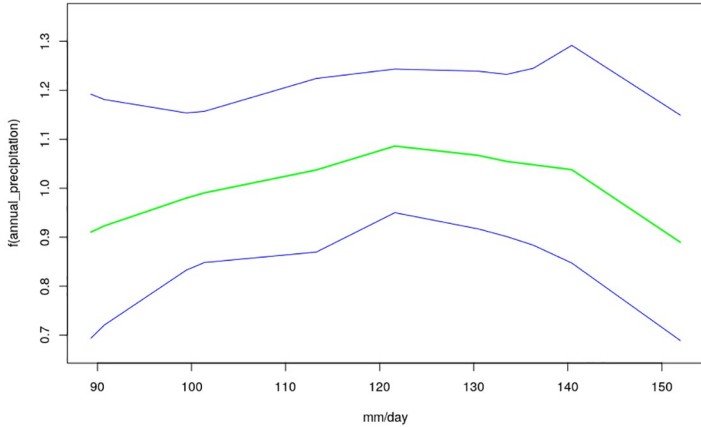

**Fig 16. Non-linear effects for the shared component of malaria and anaemia, annual precipitation (Source: Authors' creation).**

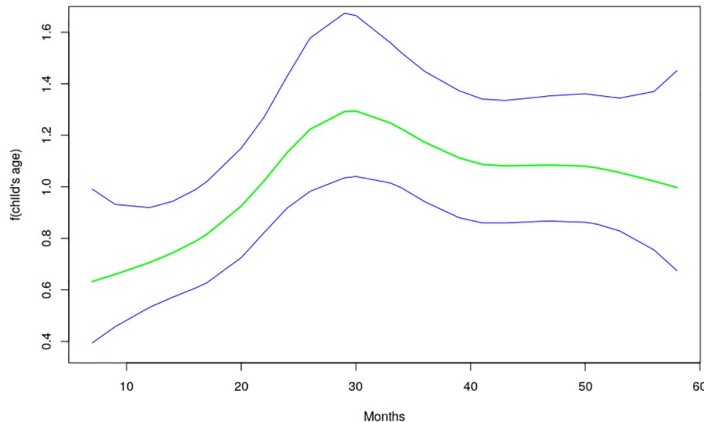

**Fig 17. Non-linear effects for malaria, child age (Source: Authors' creation).**

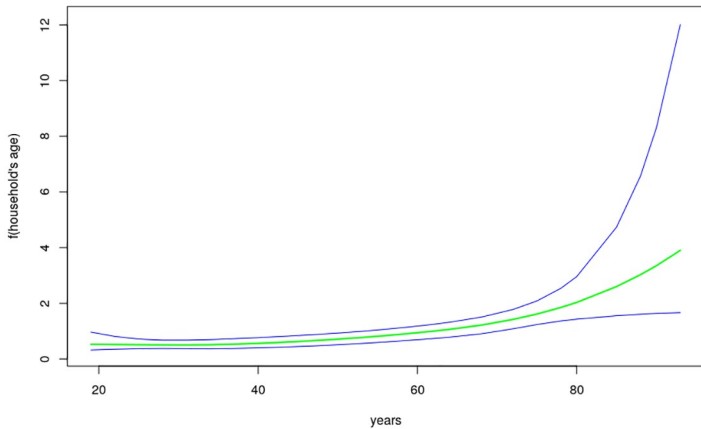

**Fig 18. Non-linear effects for malaria, household age (Source: Authors' creation).**

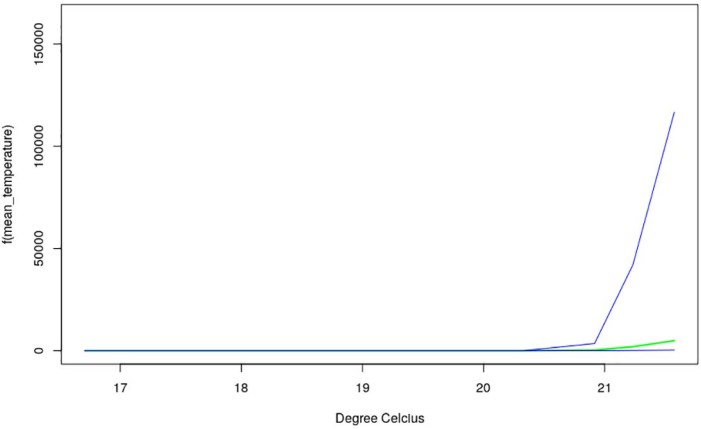

**Fig 19. Non-linear effects for malaria, mean temperature (Source: Authors' creation).**

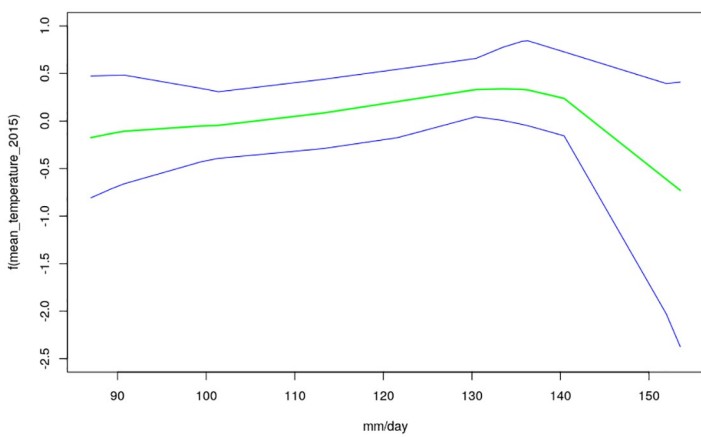

**Fig 20. Non-linear effects for malaria, annual precipitation (Source: Authors' creation).**

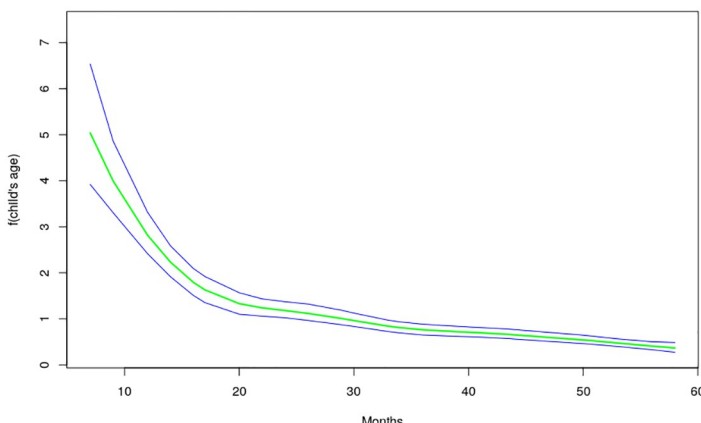

**Fig 21. Non-linear effects for anemia, child age (Source: Authors' creation).**

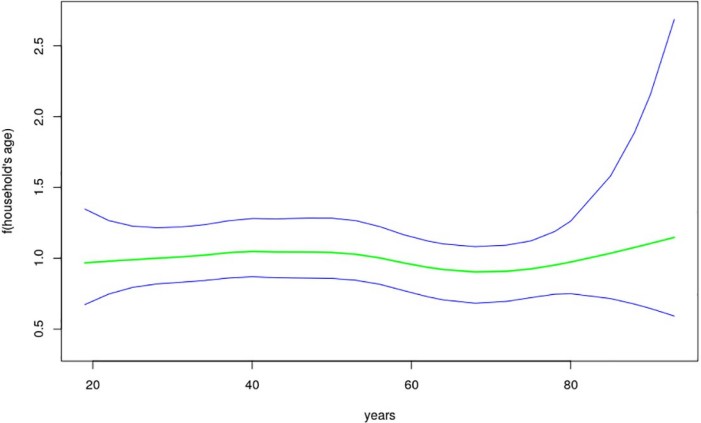

**Fig 22. Non-linear effects for anemia, household age (Source: Authors' creation).**

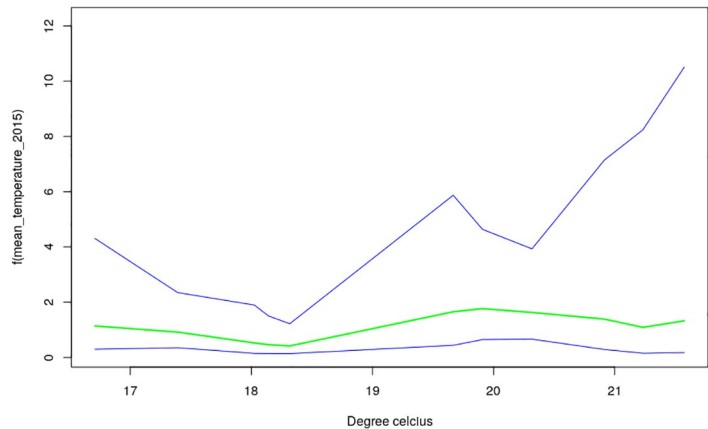

**Fig 23. Non-linear effects for anemia, mean temperature (Source: Authors' creation).**

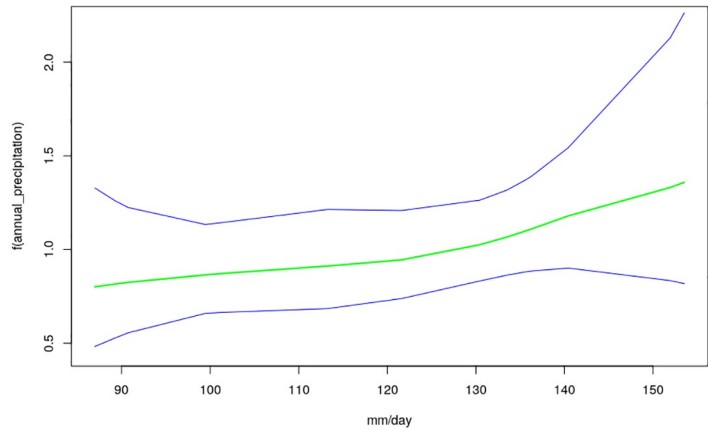

**Fig 24. Non-linear effects for anemia, annual precipitation (Source: Authors' creation).**

head of household is 75 years or older. For precipitation, the likelihood of shared risk increases with precipitation up to 120 mm/day, but starts decreasing when precipitation exceeds 120 mm/day though the credible intervals are generally wider. The findings revealed a positive association between temperature and risk of contracting both diseases, such that as temperature increases, the children become more prone to the two illnesses.

The findings for malaria Figs 17 to 20 reveal a positive association between the likelihood of having malaria and child's age up to around 30 months but beyond this age, the likelihood decreases. As for the shared pattern, the likelihood of suffering from malaria increases for children whose household heads are older than 75 years, but lower before this age. The chances of having malaria increases with precipitation below 135 mm/day, but beyond this point, the chances of having malaria decreases. As for temperature, the study findings reveal wider credible intervals reflecting higher uncertainty around the estimates thus, the estimates may need to be interpreted with caution. The non-linear effects of child's age, age of household head, annual precipitation and mean temperature estimated for anemia are presented in Figs 21 to 24. The findings show a negative association between the risk of having anemia and the child's age similar to what was obtained for the shared component, while in the case of age of household head, the risks are similar throughout the ages. Moreover, the risk of having anemia rises with increase in precipitation while the estimates for temperature appears flat but with wider credible interval.

## Discussion

A shared component modeling approach was utilized to map co-morbidity of malaria and anaemia among under-five children in Rwanda. The Bayesian spatial model allows the mapping of spatial phenomena across small geographical units and to quantify the effects of both socio-demographic and climatic factors on co-morbidity of the diseases. The findings from the study have practical relevance in a country like Rwanda where children still suffer co-morbidity from several diseases. Moreover, they provide evidence for effective and efficient implementation of intervention programs and policies designed to eliminate malaria and anaemia in Rwanda. The linear effects have shown the significant association between socio-demographic characteristics of the children and co-morbidity from the two diseases. The findings show no significant difference in the likelihood of suffering from both disease among the under-five children residing in urban and rural areas. However, findings from some African countries have shown that under-five children from urban settings are less likely to suffer from both malaria and anaemia when compared with those from rural settings [23, 24], while a Rwandan study has demonstrated lower likelihood for malaria among urban children [1]. In most rural settings in Rwanda, where more than two-thirds of the population reside, the majority of the caregivers are farmers and small scale traders who have limited access to information on disease control and prevention methods when viewed in respect to the urban dwellers. They equally suffer from acute presence of health professionals and healthcare facilities as is the case in many developing countries, and thus, overstretching the available few [1].

The findings of the study demonstrate inverse relationship between household wealth and the likelihood of co-morbidity from malaria and anaemia indicating that as wealth increases, the chances of co-morbidity reduces. These findings are consistent with thoseof [25, 26], indicating that children from impoverished households have higher prevalence of malaria compared with those from wealthy families. This may be attributed to lack of ability to afford adequate and balanced diet for children from poor families. Moreover, wealthy households would have adequate resources to acquire preventive and curative measures that safe guide their children from diseases. The similar significant association between household wealth and

likelihood of having malaria and anaemia among under-five children have been reported in several other places [2, 24].

We found that bed nets ownership reduce the risk of co-morbidity from malaria and anaemia but this was not the case for usage. Regular use of bed nets especially insecticide treated nets is a major vector control measure to prevent malaria and by extension, anemia [25]. A previous analysis of the data used in this study but where only malaria was consider revealed that both ownership and usage of bed nets reduce the risk of malaria [1]. The non-significant estimate obtained for usage in this case might have indicated that the usage of bed nets is not sufficient to cause a difference in co-morbidity from both diseases as it does for only malaria. Similar to some previous studies, we found children from male-headed household to be less prone to co-morbidity from the two diseases [26, 27]. As with most sub-Saharan African countries the majority of Rwandan households are male-headed who often provide protective leads to members of the family than what can be obtained in households headed by female. We however found no substantial difference when the sex of the child was considered.

The findings show an association between mother's educational attainment and likelihood of co-morbidity from the diseases. Children whose mothers completed at least primary education are less likely to suffer from both diseases. This association may be linked with the fact that higher educational attainment lead to improved knowledge, perception, and practice about strategies for preventing and treating diseases [28, 29]. Educated mothers are at a vintage position to know where to find and how to expressed themselves to physicians such that their children get the best of treatments than the uneducated ones will do.

The study findings revealed spatial heterogeneity in the shared risks of malaria and anaemia, and also from the individual diseases among under-five children in Rwanda. It is interesting to note that the spatial pattern obtained for malaria is similar to that observed for the shared risks of the two diseases more than does the map obtained for anemia. This finding further reinforces the synergy between the epidemiology causal pathways of malaria and anemia, which ensures that anemia is often a complication that could arise from multiple episodes of malaria [2]. Districts with substantial high risks include Rutsiro, Nyabihu, Ruhango, Gisagara, and Rusizi, while moderate risks were estimated for Rubavu, Karongi, Ngororero, Kayonza, Gatsibo, Rwamagana, Gasabo and Rulindo. The majority of the districts with high risks are places with high rates of illiteracy and poverty which would limit caregivers ability of protect their children from infectious diseases [22]. The existence of Gishwati forest, in which many agricultural and farming activities take place, could aggravate the risks for both diseases in the districts of Rutsiro and Nyabihu [30]. Also, the higher risk of co-morbidity of these diseases in Rusizi district may be compounded by the abundance of breeding sites for anopheles mosquitoes in the Nyungwe forest and Bugarama plain as reported by [31]. The high temperature often experienced in the Bugarama plain, a rice farming environment, could also increase the risk of malaria transmission by mosquitoes [32]. The presence of several agricultural activities in Ruhango and Gisagara, coupled with cross border movement can define the increased likelihood of co-morbidity of malaria and anaemia. The irrigation-based agricultural practices mainly in the wetlands of the Eastern and Southern parts of the country has been reported to influence malaria through the creation of vector breeding sites [33]. The observed spatial variation could be also linked with numerous factors including spatial disparity in human behaviour, awareness, and media exposure to malaria, child feeding practices, and disease prevention strategies. The high prevalence of malaria across the western provinces can be further explained by the presence of Lake Kivu which has been reported to speed up malaria transmission [34].

The findings established nonlinear relationships between the shared risk of the two diseases and child's age, age of household head, annual precipitation, and mean temperature, which

were considered as metrical covariates. The findings show high likelihood for the shared risks of malaria and anemia among children below 24 months of age, and this is similar to what was obtained for the specific component of anemia. Findings from other African countries have similarly exhibited this pattern of relationship [4]. The first two years of life is a period of intense growth and development, making the children to have high demand for iron, but which may not be available in the required quantity particularly for children from low socio-economic settings. Weaning food may also be obtained from some unhygienic sources which could aggravate the risk of illness and subjecting the child to anemia. Moreover, due to the fragile nature of these children, they are highly susceptible to infections.

Temperature was revealed to be positively associated with the shared risks of malaria and anaemia. An increase in temperature could shorten the time period it takes for new generations of mosquitoes to emerge, as well as the parasite's incubation period in mosquitoes [35]. Using an ecological model with malaria transmission data across African countries, [36] revealed that malaria transmission is at its optimal around temperature of $25°C$, but this decreases substantially when temperature exceeds $28°C$. The findings on annual precipitation reveals that when it is below 120 mm/day, the likelihood of co-morbidity of malaria and anemia increases with precipitation but when it is exceeds 120 mm/day, the risk decreases. Precipitation greater than 140mm destroys mosquitoes sites and leads to decrease in malaria transmission. The amount of precipitation determines the ample quantity and length of malaria transmission across different locations in a manner that in lower altitude regions, increase in precipitation might go with increase of available anopheles sites [37, 38].

This study suffers from some limitations that need to be mentioned. The study was based on district level data but this could mask fine-scale variations because the districts are composed of several spatial units. A further analysis that creates maps at a continuous scale could provide more information on local variations than is obtainable here. As with similar studies that use data from cross-sectional survey, we were unable to make causal inference from our findings. An observational study in each of the districts could provide more information regarding the pattern of infection from malaria and anemia and why they vary from place to place. Furthermore, there are possibilities of bias responses to questions that seek to elicit information on ownership and use of bed nets as the accuracy depends solely on the willingness of the respondents to provide the right response. The wealth index variable that classified the children into different wealth strata, was also computed based on households assets commonly found in urban areas whereas, most rural dwellers who could posses other wealth indicators like livestock or other farm machinery would be classified in the lowest categories resulting in mis-classification. All these notwithstanding, the data source provide adequate information that allow for spatial analysis as done here and this provide information that can enhance intervention in the country.

## Conclusion

This study aimed at understanding the shared spatial risks of malaria and anemia among under-five children in Rwanda using a shared component model while controlling for other important determinants. The study reveals that socio-demographic characteristics such as place of residence, mother's educational level, wealth index, sex of household head, bed net ownership are among the major risk factors of the shared risk of the two diseases among Rwandan children. Rising precipitation particularly above 120 mm/day is associated with decreasing risk of co-morbidity of diseases. Furthermore, the risk of developing both diseases increases as the temperature rises. Findings from the study allow for the identification of the uneven spatial variations in joint and specific risks of suffering from malaria and anemia

among young children in the country. Specifically, children from Rutsiro, Nyabihu, Ruhango, Rusizi, and Rusizi districts are particularly at higher risks. The regional variation of malaria is nearly identical to the spatial pattern of the shared risk of both diseases, indicating that malaria could the leading driver of anemia among children in the country. The study recommends considerable efforts to improve the existing intervention programs and control strategies that prioritize higher risk regions. Moreover, special attention needs to be accorded to individuals living in lowest wealth quintiles, female headed households, and children of uneducated mothers when developing malaria and anemia intervention programs and policies. Also, the study recommends that when formulating malaria and anemia intervention programs, climate and environmental factors particularly temperature and precipitation should be taken into account.

## Supporting information

**S1 Data.**
(CSV)

## Acknowledgments

PK is grateful to AIMS Rwanda for scholarship towards a master program during which the study was undertaken. The authors appreciate permission from The DHS Program to utilize the data set analysed in the study.

## Author Contributions

**Conceptualization:** Pacifique Karekezi.

**Data curation:** Pacifique Karekezi.

**Formal analysis:** Pacifique Karekezi.

**Investigation:** Pacifique Karekezi.

**Methodology:** Jean Damascene Nzabakiriraho.

**Supervision:** Ezra Gayawan.

**Validation:** Pacifique Karekezi.

**Visualization:** Pacifique Karekezi.

**Writing – review & editing:** Pacifique Karekezi.

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
