## [Decision Letter · Decision Letter 0]

11 Sep 2023

PONE-D-23-02111Shared component modeling of malaria and anemia: An estimation of the shared demographic, spatial and climatic factorsPLOS ONE

Dear Dr. Karekezi,

Thank you for submitting your manuscript to PLOS ONE. After careful consideration, we feel that it has merit but does not fully meet PLOS ONE’s publication criteria as it currently stands. Therefore, we invite you to submit a revised version of the manuscript that addresses the points raised during the review process.

We look forward to receiving your revised manuscript.

Kind regards,

Clement Ameh Yaro, Ph.D

Academic Editor

PLOS ONE

Journal Requirements:

4. We note that Figures 1-3 in your submission contain [map/satellite] images which may be copyrighted. All PLOS content is published under the Creative Commons Attribution License (CC BY 4.0), which means that the manuscript, images, and Supporting Information files will be freely available online, and any third party is permitted to access, download, copy, distribute, and use these materials in any way, even commercially, with proper attribution. For these reasons, we cannot publish previously copyrighted maps or satellite images created using proprietary data, such as Google software (Google Maps, Street View, and Earth). For more information, see our copyright guidelines: http://journals.plos.org/plosone/s/licenses-and-copyright.

a. You may seek permission from the original copyright holder of Figures 1-3 to publish the content specifically under the CC BY 4.0 license. 

5. Please remove your figures from within your manuscript file, leaving only the individual TIFF/EPS image files, uploaded separately. These will be automatically included in the reviewers’ PDF.

Reviewers' comments:

Reviewer's Responses to Questions

**Comments to the Author**

1. Is the manuscript technically sound, and do the data support the conclusions?

Reviewer #1: Yes

Reviewer #2: Yes

2. Has the statistical analysis been performed appropriately and rigorously? 

Reviewer #1: N/A

Reviewer #2: Yes

3. Have the authors made all data underlying the findings in their manuscript fully available?

Reviewer #1: Yes

Reviewer #2: Yes

4. Is the manuscript presented in an intelligible fashion and written in standard English?

Reviewer #1: Yes

Reviewer #2: Yes

5. Review Comments to the Author

Reviewer #1: REVIEW REPORT OF A PAPER ENTITLED “Shared component modeling of malaria and anemia: An estimation of the shared demographic, spatial and climatic factors”

Comment 1: Plagiarism: Similarity shows 38% and a lot came from library.nexteinstein.org (21%). Authors are advised to decrease this percentage and go below 10% at least

Comment 2: Ethical Clearance: Authors should provide sufficient information on how ethical clearance although the used data are secondary data. Who has provided that ethical clearance? What about the ethical clearance certificate Number (as an example: we used one in 2020 with the number as follows: No 235/CMHS IRB/2020)? In Rwanda ethical clearance are normally given by the College of Medicine and Health Science Institutional Review Board at the University of Rwanda.

Comment 3: Can authors add a table of socio-economic factors for interviewees?

Comment 4: What are the reasons behind this “the highest risks of contracting malaria and anemia can be found among children from Rutsiro, Nyabihu, Ruhango, Gisagara, and Rusizi districts, while those from Rubavu, Karongi, Ngororero, Kayonza, Gatsibo, Rwamagana, Gasabo and Rulindo have moderate risks of contracting both diseases”?

N.B: I do expect to read this again deeply after decreasing the Similarity percentage

Reviewer #2: This is interesting paper , which has robust methods . I commend the authors for this great piece. However, I have a few remarks for improvement ,especially in the methods section.

1. In the methods sections , kindly describe the outcome of interest and how comorbidity was generated. Also talk about the other predictors and any modifications you made, if any. You essentially need a subheading on the Dependent and independent variables.

2.The DHS uses a complex survey design. Kindly describe how you accounted for complex survey design in the analysis, both univariate and spatial, if any.

3. I am assuming the spatial shape files were used. If that was the case, kindly provide some details in the methods.

4. If your approach to the spatial mapping used the geolocation data, then in the ethics section kindly discus how and why the displacement of the GPS locations of the center points of the clusters was done in the DHS, and how this may affect the accuracy of the spatial maps ,if any.

4. Which software did you use to perform the demographics and spatial analysis. Please talk about the software used.

5. Add a section on limitations of the study.

6. In summary, kindly add more information in the methods so that the average reader can follow what was done step by step. Otherwise, great piece. I will look forward to the reading your revisions.

6. PLOS authors have the option to publish the peer review history of their article (what does this mean?). If published, this will include your full peer review and any attached files.

Reviewer #1: **Yes: **Telesphore Kabera

Reviewer #2: No

---

## [Author Response · Author response to Decision Letter 0]

11 Nov 2023

RESPONSES TO ACADEMIC EDITOR AND REVIEWERS’ COMMENTS

We thank the Academic Editor and Reviewers for reading through our manuscript and for the useful comments provided, which have helped us to improve on the quality of the manuscript. We provide below, our responses to the comments and hope that this version would meet the minimum publishable standard of PLOS ONE.

Academic editor

Comment 1

This has been done

Comment 2

Please update your submission to use the PLOS LaTeX template.

This has been done

Comment 3

We note that you have indicated that data from this study are available upon request. PLOS only allows data to be available upon request if there are legal or ethical restrictions on sharing data publicly.

There is no legal or ethical restrictions on sharing data publicly, for this time the used dataset was uploaded as a supporting information file.

Comment 4

 We note that Figures 1-3 in your submission contain [map/satellite] images which may be copyrighted. All PLOS content is published under the Creative Commons Attribution License (CC BY 4.0), which means that the manuscript, images, and Supporting Information files will be freely available online, and any third party is permitted to access, download, copy, distribute, and use these materials in any way, even commercially, with proper attribution. For these reasons, we cannot publish previously copyrighted maps or satellite images created using proprietary data, such as Google software (Google Maps, Street View, and Earth).

The uploaded figures are our own figures produced using R program; they are not map or satellite images. Due to the fact that all submitted figures are made by ourselves, no copy right is required.

 You can double-check the tiff figures we re-uploaded. 

Comment 5

Please remove your figures from within your manuscript file, leaving only the individual TIFF/EPS image files, uploaded separately. These will be automatically included in the reviewers’ PDF.

The figure were removed from the manuscript file and the tiff image files were uploaded separately.

Reviewer reports:

PONE-D-23-02111 Shared component modelling of malaria and anemia: An estimation of the shared demographic, spatial and climatic factors.

Reviewer 1

Comment 1

Plagiarism: Similarity shows 38 %. Authors were advised to decrease the percentage and go below 10 % at least.

The percentage has been decrease below 10%

Comment 2

Ethical clearance.

The study used a secondary data obtained from the Demographic and Health Survey. Data are available to anyone who wishes to conduct study, and after registering, it is permitted to utilize the data for research.

Comment 3

Can authors add a table of socio-economic factors for interviewees?

This has been done

Comment 4

The reasons behind the highest risks of contracting malaria and anemia that can be found among children from Rutsiro, Nyabihu, Ruhango, Gisagara and Rusizi districts, while those form Rubavu, Karongi, Ngororero, Kayonza, Gatsibo, Rwamagana, Gasabo and Rulindo have moderate risks of contracting both diseases.

This has been added into the manuscript.

Reviewer 2

Comment 1

In the methods sections, kindly describe the outcome of interest and how comorbidity was generated. Also talk about the other predictors and any modifications you made, if any. You essentially need a subheading on the Dependent and independent variables.

This has been done

Comment 2

The DHS uses a complex survey design. Kindly describe how you accounted for complex survey design in the analysis, both univariate and spatial, if any.

This has been done

Comment 3

I am assuming the spatial shape files were used. If that was the case, kindly provide some details in the methods.

This has been done

Comment 4

If your approach to the spatial mapping used the geolocation data, then in the ethics section kindly discus how and why the displacement of the GPS locations of the centre points of the clusters was done in the DHS, and how this may affect the accuracy of the spatial maps ,if any

This has been done

Comment 5

 Which software did you use to perform the demographics and spatial analysis? Please talk about the software used.

This has been done

Comment 6

Add a section on limitations of the study.

This has been done

Comment 7

In summary, kindly add more information in the methods so that the average reader can follow what was done step by step. Otherwise, great piece. I will look forward to the reading your revisions.

This has been done

---

## [Decision Letter · Decision Letter 1]

23 Jan 2024

Shared component modeling of malaria and anemia: An estimation of the shared demographic, spatial and climatic factors

PONE-D-23-02111R1

Dear Dr. Karekezi,

We’re pleased to inform you that your manuscript has been judged scientifically suitable for publication and will be formally accepted for publication once it meets all outstanding technical requirements.

Kind regards,

Clement Ameh Yaro, Ph.D

Academic Editor

PLOS ONE

Additional Editor Comments (optional):

Reviewers' comments:

Reviewer's Responses to Questions

**Comments to the Author**

1. If the authors have adequately addressed your comments raised in a previous round of review and you feel that this manuscript is now acceptable for publication, you may indicate that here to bypass the “Comments to the Author” section, enter your conflict of interest statement in the “Confidential to Editor” section, and submit your "Accept" recommendation.

Reviewer #1: All comments have been addressed

Reviewer #2: All comments have been addressed

2. Is the manuscript technically sound, and do the data support the conclusions?

Reviewer #1: Yes

Reviewer #2: Yes

3. Has the statistical analysis been performed appropriately and rigorously? 

Reviewer #1: Yes

Reviewer #2: Yes

4. Have the authors made all data underlying the findings in their manuscript fully available?

Reviewer #1: Yes

Reviewer #2: Yes

5. Is the manuscript presented in an intelligible fashion and written in standard English?

Reviewer #1: Yes

Reviewer #2: Yes

6. Review Comments to the Author

Reviewer #1: I am happy with your responses and as I mentioned above, all comments have been addressed. The serious issue was the high percentage of plagia.

Reviewer #2: (No Response)

7. PLOS authors have the option to publish the peer review history of their article (what does this mean?). If published, this will include your full peer review and any attached files.

Reviewer #1: **Yes: **Telesphore Kabera

Reviewer #2: No

---

## [Editor Report · Acceptance letter]

29 Mar 2024

PONE-D-23-02111R1 

PLOS ONE

Dear Dr. Karekezi, 

I'm pleased to inform you that your manuscript has been deemed suitable for publication in PLOS ONE. Congratulations! Your manuscript is now being handed over to our production team.

Kind regards, 

on behalf of

Dr. Clement Ameh Yaro 

Academic Editor

PLOS ONE